# Exercise Counteracts the Deleterious Effects of Cancer Cachexia

**DOI:** 10.3390/cancers14102512

**Published:** 2022-05-19

**Authors:** Stavroula Tsitkanou, Kevin A. Murach, Tyrone A. Washington, Nicholas P. Greene

**Affiliations:** 1Cachexia Research Laboratory, Exercise Science Research Center, Department of Health, Human Performance and Recreation, University of Arkansas, Fayetteville, AR 72701, USA; st060@uark.edu; 2Molecular Muscle Mass Regulation Laboratory, Exercise Science Research Center, Department of Health, Human Performance and Recreation, University of Arkansas, Fayetteville, AR 72701, USA; kmurach@uark.edu; 3Exercise Muscle Biology Laboratory, Exercise Science Research Center, Department of Health, Human Performance and Recreation, University of Arkansas, Fayetteville, AR 72701, USA; tawashin@uark.edu

**Keywords:** physical activity, protein turnover, inflammation, mitochondria, satellite cells, cancer comorbidities, concurrent training

## Abstract

**Simple Summary:**

This review provides an overview of the effects of exercise training on the major mechanisms related to cancer cachexia (CC). The review also discusses how cancer comorbidities can influence the ability of patients/animals with cancer to perform exercise training and what precautions should be taken when they exercise. The contribution of other factors, such as exercise modality and biological sex, to exercise effectiveness in ameliorating CC are also elaborated in the final sections. We provide meticulous evidence for how advantageous exercise training can be in patients/animals with CC at molecular and cellular levels. Finally, we emphasise what factors should be considered to optimise and personalise an exercise training program in CC.

**Abstract:**

Cancer cachexia (CC) is a multifactorial syndrome characterised by unintentional loss of body weight and muscle mass in patients with cancer. The major hallmarks associated with CC development and progression include imbalanced protein turnover, inflammatory signalling, mitochondrial dysfunction and satellite cell dysregulation. So far, there is no effective treatment to counteract muscle wasting in patients with CC. Exercise training has been proposed as a potential therapeutic approach for CC. This review provides an overview of the effects of exercise training in CC-related mechanisms as well as how factors such as cancer comorbidities, exercise modality and biological sex can influence exercise effectiveness in CC. Evidence in mice and humans suggests exercise training combats all of the hallmarks of CC. Several exercise modalities induce beneficial adaptations in patients/animals with CC, but concurrent resistance and endurance training is considered the optimal type of exercise. In the case of cancer patients presenting comorbidities, exercise training should be performed only under specific guidelines and precautions to avoid adverse effects. Observational comparison of studies in CC using different biological sex shows exercise-induced adaptations are similar between male and female patients/animals with cancer, but further studies are needed to confirm this.

## 1. Introduction

Cancer cachexia (CC) is a devastating, multifactorial and so far irreversible syndrome characterised mainly by loss of skeletal muscle mass and body fat [1]. CC affects up to 80% of cancer patients and is responsible for 20–30% of cancer deaths [1,2]. The stages of CC include: (i) precachexia, clinically defined as ≤5% weight loss without lifestyle modification over the past 6 months; (ii) cachexia, defined as >5% involuntary weight loss over the past six months or a body mass index (BMI) < 20 kg/m²; and (iii) refractory cachexia, characterised by an irreversible rapid weight loss accompanied by a life expectancy < three months [3]. 

The major known mechanisms contributing to the development and progression of CC include imbalanced protein turnover, inflammatory signalling, mitochondrial degeneration and muscle stem cell (satellite cell) dysregulation [4,5]. Specifically, CC is characterised by increased muscle protein degradation combined with decreased muscle protein synthesis [5]. Activation of protein degradation signalling through either the ubiquitin proteasome system (UPS) or the lysosomal proteasome system (LPS) is well described in different preclinical and clinical models [5,6,7,8,9,10]. Chronic inflammation characterised by increased levels of inflammatory factors in skeletal muscle such as tumour necrosis factor α (TNF-α) and IL-6 is also a major mechanism of CC in both patients and mouse models [11,12,13]. Furthermore, mitochondrial degeneration including network degeneration, elevated levels of reactive oxygen species (ROS) production/emission, decreased mitochondrial quality and reduced aerobic metabolism [5,14,15] are considered hallmarks of CC. Finally, skeletal muscle satellite cells, as well as their regulators known as myogenic regulatory factors, may play a role in mechanisms that underlie CC [16]. Satellite cell differentiation is inhibited in CC and muscle fibres that are damaged during CC may undergo degeneration [4]. The contributions of dysregulated protein turnover, increased chronic inflammation, mitochondrial dysfunction and myogenic disruption to the development and progression of CC are complex and intertwined. Future therapeutic interventions for CC should be focused on combating this constellation of dysfunction. 

Nutritional interventions are insufficient to fully reverse the loss of body weight induced by CC [17,18]. To date, there is no effective treatment to counteract wasting in patients with CC. Exercise training is considered a therapeutic approach for many chronic diseases. Generally, exercise can improve cardiorespiratory and neuromuscular systems and promote psychological wellbeing [19,20,21]. The positive effects of exercise are accompanied by beneficial adaptations at a molecular level [22]. Specifically, exercise training increases the gene expression of skeletal muscle myogenic factors (e.g., Pax7, MyoD, Myogenin etc.) [23] and activates molecular pathways that regulate skeletal muscle mitochondrial biogenesis (e.g., elevated levels of PGC1-α) [24]. Furthermore, systematic exercise training reduces chronic circulating inflammatory IL-6 levels, especially in individuals with a non-active lifestyle [25]. 

Exercise training has been proposed as a non-pharmacological therapeutic approach for CC [26,27,28]. The potential of exercise training to decrease cancer-induced inflammation and oxidative stress [29], as well as mitigate cancer-induced suppression of anabolic signalling and protein synthesis [30] can be considered an efficient “tool” to attenuate skeletal muscle abnormalities observed in CC. Preclinical studies highlight the benefits of exercise training in animals with different types of cancer (e.g., *Apc^Min/+^* and C26 colorectal cancer mice [31,32], breast cancer mice [33,34,35,36], SENCAR skin cancer mice [37]) and rat models such as Walker 256 carcinosarcoma [38,39,40]). Irrespective of the model, exercise training further reduces tumour incidence, tumour multiplicity and tumour growth [41], as well as attenuates the progression of CC [31,32]. A major unavoidable limitation of pre-clinical studies in CC mouse models is the short timeline (three to four weeks) the mice have after the inoculation of cancer cells. For this reason, the majority of pre-clinical studies begin the exercise intervention before or just after the inoculation of cancer cells to succeed exercise-induced adaptations [42]. This timepoint is undoubtedly earlier than the development and progression of CC (around three to four weeks after cancer cells inoculation). For this reason, most pre-clinical studies are designed with prevention of cachexia as the primary goal leaving a relative dearth in data regarding reversal of existent cachexia.

Previous clinical studies in patients with cancer from early to advanced stages of disease have concluded that exercise training, independent of modality (i.e., resistance, endurance or concurrent training), has beneficial effects on skeletal muscle health, physical performance, quality of life and psychological health [27,28,43,44,45,46]. Exercise training can be also advantageous as a pre-operative intervention in cancer patients [47,48,49]. Specifically, exercise not only improves the physical capacity and function of cancer patients [47,48], but also ameliorates postoperative recovery [47] and may have a protective direct effect on tumour growth through altering the gene expression of immunity and inflammation “key” factors [49]. Given the heterogeneity observed in cancer patients (i.e., different cancer diagnoses, cancer stages and symptoms) in exercise clinical trials, further works are necessary to evaluate exercise training as a monotherapy or combinatorial therapy throughout all phases of cancer, including the phases of CC development and progression [42,50,51]. To this point, ongoing clinical trials with exercise-based interventions in cancer patients are presented in the new Appendix A.

Unfortunately, exercise training is not always feasible in humans or animals with cancer due to cancer- and chemotherapy-induced fatigue, anaemia, cardiac dysfunction and other comorbidities [52,53,54]. For instance, mild endurance training in C26 colorectal cancer mice, a CC mouse model that suffers additionally from anaemia, does not prevent body weight and muscle loss, and even worsened their condition [53,55]. In cases of comorbidities along with cancer, exercise training should be performed only under specific guidelines, precautions and supervision by specialists. The exercise workload including exercise intensity, duration, frequency and modality should be meticulously assessed in each case individually. Further preclinical studies using a translatable exercise framework that can account for the comorbidities accompanying cancer are needed to optimise exercise training programs to avoid any potential exercise-induced adverse effect. Although all forms of exercise (resistance, endurance and concurrent training) induce beneficial adaptations in CC-induced skeletal muscle damage [30,31,32,56,57], concurrent resistance and endurance training may surpass resistance training *per se* or endurance training *per se*. Such an approach improves both anaerobic and aerobic skeletal muscle metabolism simultaneously [58,59,60], both of which are compromised with CC. 

Biological sex differences have been observed in CC development and progression, with females more resistant to the cancer-induced skeletal muscle decline [15,61]. However, no study has directly investigated if exercise-induced beneficial adaptations in skeletal muscle of patients or mice with CC differ between females and males. Studies in healthy individuals have shown females and males respond differently to exercise stimuli [62,63]. When observationally comparing studies using the same CC mouse model but different biological sex, exercise-induced adaptations are similar in females and males [57,64]. Further investigation is needed to confirm these observations. 

The primary purpose of this review is to highlight the effects of exercise training on molecular and physiological mechanisms related to CC, namely protein turnover, inflammatory signalling, mitochondrial function and satellite cell-related process. Importantly, we emphasise the contraindications of exercise in some cancer conditions with comorbidities. Secondly, we describe the effects and importance of different exercise modalities. Based on the current literature, we propose the most advantageous exercise modality for cancer patients. Finally, we aim to specify the differences between females and males in response to exercise training during CC development and progression.

## 2. Effects of Exercise Training on Cancer Cachexia (CC) Mechanisms

### 2.1. Protein Turnover in Response to Exercise

One of the major hallmark features of CC is the net loss of skeletal muscle protein due to the imbalance between protein breakdown and protein synthesis [5]. Preclinical studies in Lewis Lung Carcinoma (LLC) male [7] and female [15] mice, as well as in a colon cancer cachectic mouse model, *Apc^Min/+^* mice [6] revealed a combination of a loss of protein synthesis and increased markers of protein breakdown. Specifically, protein fractional synthesis rate (FSR) decreases around 40%, while muscle protein ubiquitination and protein levels of ubiquitin-proteasome marker FOXO1 increase around 50% in male LLC mice 4 weeks after tumour implantation [7]. Similar to males, cachectic female LLC mice also present reduced muscle protein FSR in gastrocnemius, whereas the mRNA levels of Ubiquitin C, Atrogin-1 and Murf-1 are elevated [15]. In accordance with the results in LLC mice [7,15], myofibrillar protein synthesis as well as mRNA content of IGF-1 and phosphorylation of the mTOR signalling cascade decrease during CC, while protein degradation (ATP dependent and independent) increases during CC progression in *Apc^Min/+^* mice [6]. For further information about the advantages and disadvantages of preclinical CC mouse models, the reader is directed to a recent review highlighting this topic [5].

Exercise training increases protein synthesis in both healthy and diseased populations [65,66,67], while decelerating the high protein degradation induced by chronic diseases [68,69]. Improvements in protein synthesis have also been observed in different cancer mouse models after performing exercise training [30,70,71]. Specifically, endurance-type wheel running (60 min in 5–6.5 m/min, 5 days/week, 11 weeks) leads to enhanced phosphorylation of mTOR, a critical regulator of skeletal muscle protein synthesis [72], in female C26-bearing mice [71]. Repeated eccentric contractions (10 tetanic contractions in 100 Hz, 6–12 V × 6 sets, 4 times/week for 2 weeks) of the tibialis anterior (TA), considered a representative model of resistance exercise training for animal studies, improves protein synthesis and mTORC1 signalling and attenuates atrophy of oxidative and glycolytic muscle fibres in male *Apc^Min/+^* mice [30]. Activation of protein synthesis and mTORC1 signalling is also induced by a single bout of eccentric contractions in male *Apc^Min/+^* mice. Acute activation of mTORC1 occurs on the condition that the electrical stimulation that elicits the eccentric contractions is not low-frequency (i.e., 10 Hz, 5 V) [73] but is sufficient to induce tetanic contractions (i.e., 100 Hz, 6–12 V) [70]. Performing either single [64] or multiple bouts of tetanic eccentric contractions [57] in female [64] and male [57] *Apc^Min/+^* mice not only activates protein synthesis signalling but also suppresses muscle catabolic signalling. Protein synthetic responses occur by reducing the protein expression of cellular energy-sensing enzyme 5′-adenosine monophosphate-activated protein kinase (AMPK), which is an established mTOR inhibitor and chronically activated in severely cachectic muscle [74]. 

Beyond anabolic signalling, AMPK is known to activate catabolic signalling, in addition to its canonical functions in promoting energy metabolism and mitochondrial adaptation [75]. Herein, inhibition of AMPK is also observed after 8-week moderate-intensity treadmill running (1 h at 18 m/min, 5% grade, 6 days/week) in male *Apc^Min/+^* mice [31]. Furthermore, voluntary wheel running (for ~2.5 weeks) counteracts cancer-induced protein degradation through suppressing both the induction of ubiquitin ligases (*Atrogin1*, also known as *Fbxo32*; and *Murf1*, also known as *Trim63*) and the protein expression of autophagic markers (LC3bII/LC3bI ratio and p62/Gapdh ratio) in female C26 tumour-bearing mice [32]. Similarly, decreased LC3bII/LC3bI ratio is also observed by either endurance training (15–45 min wheel running at 5–11 m/minor, 3 days/week, 2.5 weeks) [76] or concurrent resistance (inclined ladder climbing with gradually increased resistance load) and aerobic (25-min wheel running in 5–9 m/min speed) training (4 days/week, 5.5 weeks) [58] in male C26 tumour-bearing mice. Interestingly, similar exercise adaptations in skeletal muscle protein turnover are observed when tumour-bearing mice are treated with chemotherapy agents [77]. Specifically, moderate exercise training on a treadmill (40–60 min at 60% of maximum speed, 5 days/week, 3 weeks) reduces the gene expression of catabolic markers (*Fbxo32*, *Trim63* and *Myostatin*) related to atrophy and increases it in LLC mice treated with doxorubicin chemotherapy drug [77]. 

Similarly, a clinical trial reports that 10-week concurrent resistance (knee extension, leg press, lateral pull-down, chest-press, back extension and sit-ups) and endurance (ergometer cycling) training in cancer patients treated with chemotherapy can prevent cancer- and chemotherapy-induced disruptions in molecular signalling cascades associated with the ubiquitin proteasome system (e.g., Atrogin-1 and Murf-1) and protein synthesis (e.g., mTOR) [78]. As a result, exercise training seems to be a good strategy to combat unbalanced protein turnover induced by cancer and chemotherapy in preclinical models and clinical applications (Figure 1 and Table 1); however, further clinical trials are needed to establish this empirically.

### 2.2. Inflammatory Signaling in Response to Exercise 

Chronic inflammation accompanied by elevated circulating inflammatory cytokines is another major hallmark mechanism of CC pathogenesis [11]. Physical activity is associated with lower odds of having elevated inflammation levels (e.g., elevated levels of C-reactive protein—CRP) [79,80,81]. In preclinical studies, there is evidence that exercise has an anti-inflammatory effect by reducing TNF-α expression [82,83,84]. The mechanism underlying the anti-inflammatory response after performing exercise training is related to the major role of the cytokine IL-6 [81]. Acute exercise can increase circulating levels of IL-6 [85], which induces the production of the anti-inflammatory cytokines IL-1ra and IL-10 [85,86]. Exercise also inhibits the production of the inflammatory cytokine, TNF-α [81,87,88]. Acute elevation of IL-6 after exercise stimulus is therefore considered a beneficial response that could promote protein synthesis and advantageous skeletal muscle adaptations [89]. On the other hand, long-lasting elevated systemic IL-6 levels activate catabolic signalling pathways associated with muscle wasting [89]. Long-term exercise training decreases circulating IL-6 levels, especially in individuals following a sedentary lifestyle [25]. Also, an inverse correlation between the amount of weekly physical activity and the level of plasma IL-6 in the resting state occurs in healthy males [90]. In addition, aged lifelong exercisers with an aerobic training history of ~50 years not only present with lower serum IL-6 levels compared to old healthy non-exercisers in the resting state, but also have similar post-exercise (4 h after an acute resistance training session) gene expression of skeletal muscle anti-inflammatory markers compared to young exercisers [91].

The anti-inflammatory benefits of exercise training have been described in preclinical and clinical cancer studies [33,92,93,94,95]. Specifically, exercise training by treadmill (60 min at 20 m/min, 5% grade, 6 days/week, 20 weeks) reduces the plasma concentration of two major inflammatory markers, MCP-1 and IL-6, in female breast-tumour bearing mice (C3(1)SV40Tag mouse model) [33]. This inflammatory reduction is associated with the deceleration of breast tumour progression observed in C3(1)SV40Tag mice after the 20-week exercise training program [33]. Treadmill running (15–30 min at 15–20 m/min, 3 days/week, 6 weeks) also reduces TNF-α concentration in the plasma and TNF-α gene expression in the colon of mice with azoxymethane (AOM)-induced colon cancer [92]. Similarly, voluntary wheel running for 4 weeks in breast cancer mice (PyMT) decreases not only circulating but intramuscular levels of TNF-α protein, as well as reduces the intramuscular mRNA content of several TNF-α target genes (Traf2, IκB-α, Ank1 and NFκB1) [36]. Furthermore, a treadmill running program (25–40 min at 14–20 m/min, 5 days/week, 8–14 weeks), starting either before or after injecting estrogen-dependent MC4L2 cancer cells in female mice, reduces breast tumour volume and the levels of IL-6 in the tumour, on condition that the mice continue exercising after the onset of tumorigenesis [93]. In addition, the increased protein concentration of IL-6 observed in tumour of LLC mice treated with doxorubicin chemotherapy is mitigated by 3-weeks of moderate exercise training on a treadmill (40–60 min at 60% of maximum speed, 5 days/week) [77].

Surprisingly, a preclinical study in *Apc^Min/+^* colorectal cancer mice showed that treadmill running (1 h at 18 m/min speed and 5% grade, 6 days/week, 8 weeks) does not suppress systemic IL-6 overexpression induced by electroporating IL-6 plasmid in quadriceps. This result is surprising since an exercise training program effectively prevents the IL-6-induced decrease in body weight of male *Apc^Min/+^* mice [31]. Although exercise training improves phenotypic characteristics in male *Apc^Min/+^* mice, such as lean mass and motor function, 5-week voluntary wheel running does not decrease either tumour burden or the elevated circulating IL-6 levels [96].

When data from two randomised controlled exercise training trials in breast cancer patients were pooled and analysed together, the findings showed that exercise training (12–18 weeks of resistance or concurrent training) does not decrease the cancer/chemotherapy-induced levels of the inflammatory markers IL-6 and IL-6/IL-1ra ratio [97]. However, it seems that exercise intensity plays a crucial role on the inflammatory profile of patients with cancer during and after cancer treatment (chemotherapy and/or radiotherapy and/or endocrine therapy) [98]. Specifically, performing high-intensity concurrent resistance and aerobic training during cancer treatment results in a lesser increase of the plasma inflammatory markers CRP and TNF-α immediately after completion of cancer treatment, compared to performing low-intensity concurrent resistance and aerobic training [98]. Also, many clinical trials have concluded that exercise training promotes an anti-inflammatory profile in cancer patients undergoing chemotherapy or radiation therapy [94,99,100]. Specifically, combination of resistance training with high-intensity aerobic training (60-min sessions, 2 days/week for 16 weeks) [100] or resistance training *per se* (60-min sessions, 2 days/week, 12 weeks) [99] mitigates [100] and even counteracts [99] cancer-mediated increased levels of IL-6. In conclusion, exercise training suppresses the CC-induced elevation of inflammatory markers in different CC mouse models except for *Apc^Min/+^* mice, as well as in humans, but only on condition that exercise intensity is sufficient (Figure 1 and Table 1). Although concurrent resistance and endurance training is considered the optimal exercise mode in cancer populations, further investigation is needed to understand what intensity and duration of exercise training are ideal to promote an anti-inflammatory profile in cancer patients.

### 2.3. Mitochondrial Function and Health in Response to Exercise

As observed in many pathologies, mitochondrial function and health declines with CC. Mitochondrial dysfunction includes signalling pathways and mechanisms of mitochondrial biogenesis, fusion and fission [14,101,102], production of reactive oxygen species (ROS) [14,103,104], as well as respiratory function [14,105], all of which influence CC development and progression [5]. Results from our laboratory show that mitochondrial degeneration precedes muscle wasting in tumour-bearing mice [14]. Specifically, increased mitochondrial ROS production (one week after cancer cell implantation), degeneration of the mitochondrial network (two weeks after implantation) and impaired respiratory function (three weeks after implantation) were observed prior to muscle mass loss (4 weeks after implantation) in LLC male mice [14]. These results accompanied by previous findings [101,102] suggest mitochondrial degeneration as a potential key promoter of CC and, as such, mitochondrial quality as a potential therapeutic target for the treatment of CC.

It is well-established that exercise training improves mitochondrial health inducing robust adaptations in both mitochondrial content and quality [106,107]. Specifically, exercise training promotes oxidative phosphorylation and respiratory capacity, promotes mitochondrial biogenesis and decreases ROS emission [106,107,108]. Mitochondrial function and health are also improved with exercise training in patients and rodents with cancer [55,56,58,109,110]. Specifically, concurrent resistance (inclined ladder climbing with gradually increased resistance load) and aerobic (25-min wheel running in 5–9 m/min speed) training (4 days/week, 5.5 weeks) in male C26 tumour-bearing mice may preserve mitochondrial function as indicated by preventing the reduction in muscle succinate dehydrogenase (SDH) activity induced by tumour growth [58]. Although it was not statistically significant, a trend toward increased PGC-1α levels was also observed in the C26 mice that performed concurrent training [58]. Interestingly, significant increases in the protein and gene expression levels of PGC-1α, the master regulator of mitochondrial biogenesis [111], have been noted in C26 and LLC tumour-bearing mice when the exercise intervention (45 min running on treadmill at 14 m/min, 5 days/week for 2–8 weeks) is combined with erythropoietin (EPO) treatment for preventing anaemia [55]. Anaemia is identified in 40% to 80% of patients who have cancer [112] as well as in different cancer mouse models [113] and impairs oxidative metabolism due to hypoxia [114]. A recent study found that 4-week voluntary wheel running effectively counteracted mitochondrial dysfunction preventing muscle mass loss in C26 mice, without compromising running activity [109]; this occurred even though an EPO treatment was not administered. Specifically, the voluntary wheel running attenuated CC-induced mitochondrial abnormalities in tumour-bearing C26 mice attenuating: (i) decreased expression levels of mitochondrial-related proteins (OXPHOS subunit proteins and PGC-1α); (ii) decreased mitochondrial enzyme activity (citrate synthase and cytochrome c oxidase); (iii) decreased expression levels of mitochondrial fusion and fission proteins (Mfn2 and Drp1); (iv) increased mitochondrial oxidative stress; and (v) increased appearance of damaged mitochondria with disrupted cristae structure [109]. In addition, five-week voluntary wheel running increases the protein expression of mitochondrial complex II and IV in male *Apc^Min/+^* mice [96].

The beneficial effect of exercise training on mitochondrial health and function in CC mice has been confirmed by a clinical trial in women with breast cancer during chemotherapy [56]. A 16-week training program (2 days/week), including either resistance training combined with high-intensity interval training or aerobic training combined with high-intensity interval training, increased mitochondrial content in the skeletal muscle of cancer patients, as indicated by the elevated levels of citrate synthase activity [56]. However, compared to controls (cancer patients treated with usual care), increased levels of the electron transport chain proteins (complex I, II and IV) were observed only in patients that performed concurrent aerobic training and high-intensity interval training, but not in patients that performed concurrent resistance and high-intensity interval training [56]. Therefore, the training volume as well as the aerobic nature of the exercise stimulus are two major factors to successfully preserve mitochondrial function in cancer patients undergoing chemotherapy. In conclusion, exercise training improves mitochondrial health and function in skeletal muscle of mice and patients with cancer (Figure 1 and Table 1), with aerobic training being the optimal exercise stimulus that favours mitochondrial adaptations.

### 2.4. Satellite Cells and Myogenic Regulatory Factors in Response to Exercise

Skeletal muscle satellite cells, as well as their regulators known as myogenic regulatory factors (MRFs; e.g., Myf5, MyoD, myogenin and MRF4) and Pax7 play a role in mechanisms underlying CC [16]. Satellite cells (SC), along with other progenitor populations, become activated and enter a regenerative program in response to cancer-induced skeletal muscle membrane damage in both cancer patients and tumour-bearing mouse models [4]. Muscle membrane damage is reportedly mediated by excessive systemic inflammation from tumours. Due to Pax7 overexpression, myogenic cells are unable to fuse and help alleviate muscle fibre damage [4]. Inaba et al. [115] concluded that the proliferation and differentiation abilities of muscle stem cells derived from the C26 tumour-bearing mice are sustained in vitro. However, inefficient regeneration process in skeletal muscle in C26 tumour-bearing mice, as determined by immunohistological analysis (eMyHC staining), is observed in vivo, potentially due to the cancer-induced decrease of neutrophils, macrophages, and mesenchymal progenitors [115]. While speculative, impaired satellite cell function during CC could also potentially mediate muscle atrophy due to dysregulation of fusion-independent communication to muscle fibres and/or interstitial cells [116,117,118,119]. In addition, chronic doxorubicin (DOX) administration, a highly effective chemotherapeutic agent for treating various types of cancer, reduces SC density in the soleus of ovariectomised female rats [120]. Considering that cancer *per se* affects satellite cell behaviour, the combination of both the disease and chemotherapies may induce SC dysfunction and exacerbate CC [121].

Exercise training can promote hypertrophy and satellite cell contribution to muscle fibres through SC activation, proliferation and differentiation [122], as well as fusion-independent mechanisms [116,117,118,119]. Long-term [123] or short-term [124,125] exercise training increases the number of SCs in both healthy and diseased populations. Preclinical studies in CC mice also confirm the beneficial effects of exercise training on SC function [126]. Specifically, voluntary wheel running for 19 days normalises cancer-induced dysregulated protein expression of Pax7 to control levels, as well as restores muscle mass by increasing the size of glycolytic muscle fibres in C26 mice [126]. Resistance exercise (ladder climbing with resistance loading increased 10% bi-weekly, 3 repetitions × 5 sets, 3 days/week, 11 weeks) potentially activates myogenic cell activity in C26 mice, as indicated by increased mRNA levels of myogenin [71]. Conversely, aerobic training (60 min wheel running at 5–7 m/min speed, 5 days/week, 11 weeks) does not significantly increase the levels of any myogenic regulatory factor [71].

A clinical trial in female cancer patients treated with chemotherapy concluded the number of SCs does not change in either Type I or Type II muscle fibres after a 10-week concurrent resistance and aerobic training program [78]. However, cautious interpretation of these findings is needed since methodological limitations, including the small heterogeneous sample size of cancer patients, the different chemotherapies used for each patient and the lack of non-exercise control group, are present in this study [78]. Specifically, as a non-exercise control group is not included in this study [78], no change in SC content after exercise training could be potentially interpreted as preservation of SCs, which is a beneficial response. In agreement with the aforementioned results, resistance training (3 days/week for 9–16 weeks) does not change the SC content in either Type I or Type II muscle fibres in patients with prostate cancer undergoing androgen deprivation therapy [127] or in patients with germ cell cancer undergoing cisplatin-based chemotherapy [128]. Similarly to the exercised cancer patients, the SC content in either Type I or Type II muscle fibres does not change in the non-exercise cancer patients (control group) of both clinical trials [127,128]. However, it is worth mentioning that even if the SC content is unchanged, that does not mean the function of the SCs is unchanged as well. Nevertheless, the function of SCs, including SC activation, proliferation and differentiation, was not assessed in the aforementioned studies. In addition, to interpret exercise-induced responses, exercise workload including intensity, duration and frequency, should be considered. It is conceivable the exercise training programs of published trials [78,127,128] may not be sufficiently rigorous to induce significant changes at least at a cellular level.

In contrast to single exercise mode studies, concurrent resistance and high-intensity interval training by cycle ergometry (2 days/week for 16 weeks) increases the SC number per fibre in women with breast cancer during chemotherapy [56]. Also, the increase in SC number is significantly associated with the increase in muscle fibre CSA and muscle strength [56]. Surprisingly, in the same clinical trial no improvement of SC content was found in breast cancer patients either in the group performing 16-week concurrent aerobic and high-intensity interval training or in the usual care group (control group) [56]. Therefore, the improvement of SC content after concurrent resistance and high-intensity interval training in the presence of cancer may reflect an activation of the skeletal muscle repair process to counteract CC [56]. Exercise modality seems to play a critical role on triggering skeletal muscle SC activation. In conclusion, exercise may improve SC function in mice with cancer, but there is no clinical trial which has assessed intrinsic SC function in patients with cancer after performing exercise training. Clinical trials report an increase in the number of SCs only when patients with cancer follow concurrent resistance and high-intensity interval training by cycle ergometry (Figure 1 and Table 1). To better understand if exercise training can ameliorate satellite cell dysfunction in cancer, future studies are needed to assess SC content and SC function, as two distinct factors, that both contribute to the CC development and progression. Further attention should also be paid to the muscle microenvironment (e.g., extracellular matrix composition, inflammatory milieu, behaviour of the muscle fibre, etc.) during CC, as the satellite cell niche strongly influences satellite cell behaviour [129]. In addition, given that SCs not only contribute to muscle repair and growth but also communicate throughout muscle and specifically fibrogenic cells [116,117,118,119], it would be interesting for future studies to investigate if exercise training could decrease cancer-induced fibrosis [130] through SC-mediated mechanisms.

**Figure 1 cancers-14-02512-f001:**
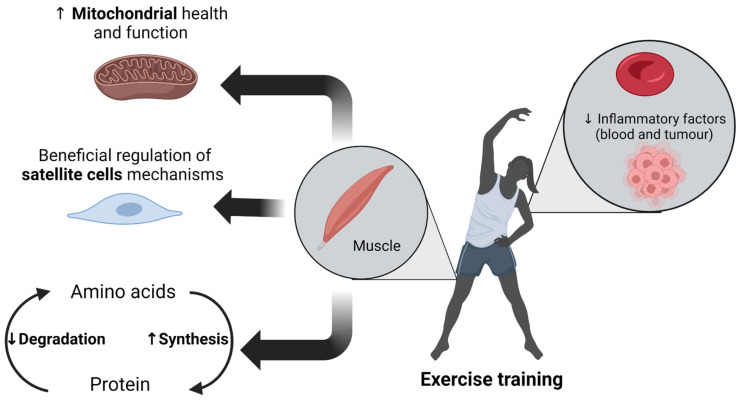
Beneficial effects of exercise training on counteracting the deleterious mechanisms of cancer cachexia (CC).

**Table 1 cancers-14-02512-t001:** Effects of exercise training on cancer cachexia (CC) mechanisms.

Protein Turnover	Inflammatory Signalling	Mitochondrial Function	Satellite Cells (SC) and Myogenic Regulatory Factors
**↑ Protein synthesis** [30,70,71,78] **↓ Protein degradation:** ↓ Ubiquitin ligases: Atrogin-1 and Murf-1 [32,77] ↓ Autophagic markers: LC3bII/LC3bI and p62/Gapdh ratio [32,58,76] ↓ AMPK [31,57,64]	↓ **Inflammatory markers**: IL-6, TNF-α [33,36,92,93,94,99,100]	↑ **Mitochondrial health and function**: ↑ PGC-1α protein and mRNA levels [55] ↑ SDH activity [58] ↑ Mitochondrial content [56] ↑ OXPHOS subunit protein levels [96,109] ↑ Mitochondrial enzyme activity [56,109] ↑ Mitochondrial fusion and fission [109] ↓ Damaged mitochondria with disrupted cristae structure [109] ↓ Mitochondrial oxidative stress [109]	↔ SC content [78,127,128] with resistance or endurance training ↑ SC content with concurrent resistance and high-intensity interval training [56] Beneficial regulation of myogenic regulatory factors: ↓ Pax7 [126] ↑ Myogenin [71]

Statistically significant increase (↑), decrease (↓) in mice/patients with cancer compared to non-exercising tumour-bearing mice or usual care treated patients with cancer or pre-training baseline results of the exercised patients with cancer. No changes (↔).

## 3. Contraindications to Exercise in Cancer

Co-existent health problems usually accompany cancer, complicating the way exercise prescription should be applied to cancer patients. Comorbidities are present in around 70% of cancer patients and 35% of these individuals have more than two comorbid conditions. The frequency of comorbidities may decrease or increase based on the type of cancer, age, gender, lifestyle and socioeconomic status of cancer patients [131,132]. The most common comorbidities associated with cancer are anaemia, cardiac dysfunction, chronic fatigue, hypertension, diabetes, asthma and arthritis [53,133,134]. Exercise is considered an unquestionably beneficial intervention to mitigate not only CC but also impairments caused by different comorbidities. However, in case of cancer being accompanied by a comorbidity, components of exercise training workload (i.e., duration, intensity, weekly frequency, rest between sets) should be carefully selected and adjusted based on every individual’s condition, so that any potential exercise-induced adverse effect can be avoided. If a comorbidity compromises the health condition of cancer patient, exercise training should not be performed until addressing the symptoms and/or causes of comorbidity.

As an example, avoidance of exercise should be recommended in anaemic cancer patients until their haemoglobin reaches normal levels [53,135]. Anaemia (haemoglobin level < 12 g/dL) occurs in approximately 40–80% of cancer patients depending on cancer stage and type [112] and may contribute to muscle wasting of cachectic cancer patients [136]. Anaemia is also a major side effect of chemotherapy [137], which may further worsen the already low haemoglobin levels induced by cancer. As mentioned before, in C26 tumour-bearing mice, in which CC is associated with anaemia, treadmill running (45 min at 14 m/min speed, 5 days/week, 8 weeks) does not improve skeletal muscle wasting, but even worsens it when the exercise training is performed for only 2 weeks [55]. However, when haemoglobin levels are corrected through EPO administration, treadmill running (45 min at 14 m/min speed, 5 days/week, 2 weeks) is able to counteract both the oxidative myofiber atrophy and the shift from oxidative to glycolytic fibre type, potentially through stimulating PGC-1α expression [55]. Moreover, in LLC mice where anaemia is more severe compared to C26 mice, exercise training *per se* (45 min treadmill running at 14 m/min speed, 5 days/week, 4 weeks) can prevent muscle strength loss. A combination of exercise training with EPO administration can additionally promote muscle oxidative capacity and intracellular ATP content, alleviate muscle wasting, and prevent the onset of mitochondrial ultrastructural alterations [55]. These results [55] are meaningful to highlight the major role of preserving haemoglobin concentration close to normal levels to prevent CC through exercise training.

The administration of EPO in cancer patients with anaemia revolutionised the treatment of this complication around 20 years ago with impressive results from a haematological perspective while simultaneously avoiding the adverse effects of blood transfusions in anaemic patients with cancer [138,139]. However, later scientific evidence shows that EPO treatment in cancer patients mediates a variety of serious adverse effects, including survival decrease [140], acceleration of tumour progression [141] and an increase in the incidence of thromboembolic events [142]. For this reason, utilisation of EPO in cancer patients has been restricted, especially among patients treated with a curative intent and in patients with advanced tumours but long-term survival expectations [143]. Iron administration (oral or intravenous) is now considered a promising alternative to normalise haemoglobin levels in anaemic cancer patients [144,145].

Heart function is also impaired in tumour-bearing mice and rats (C26 carcinoma tumour-bearing mice [146,147,148], LLC tumour-bearing mice [149], breast tumour-bearing mice [150] and liver tumour-bearing rats [151]). Also, a series of epidemiological studies conclude that there is a comorbidity of cancer and cardiovascular diseases [152,153]. Many cancer therapies (e.g., anthracyclines, HER-2-targeted agents, mitotic inhibitors, immune modulators, radiation etc.) induce cardiac toxicity and lead to long-term and most likely permanent myocardial injury [154,155]. Considering that a remarkable percentage of cancer patients additionally suffer from heart failure [156,157], further precautions should be taken when exercise is performed in these populations. Any potential adverse effect of exercise in cancer patients with a comorbidity of heart failure could be prevented by following the exercise training guidelines specific to patients with heart failure [158]. Also, in case of immunosuppressed or bone marrow transplanted cancer patients, exercise training should be avoided in public gyms until their white blood cell count returns to normal levels [135]. In addition, due to the additional energetic demands caused by the tumour, especially when cachexia has developed, appropriate nutritional support is required for patients with cancer perform exercise training [159].

Exercise training, even if it is high intensity, is considered safe and effective in cancer patients either shortly after completion of primary cancer treatment [160] or during cancer treatment [161], but on condition that no medical contraindications are present [161]. In case of cancer patients presenting comorbidities, exercise training should be performed only under specific guidelines, precautions and supervision by specialists. Future preclinical exercise studies should consider the limitations of human exercise oncology studies related to cancer- and treatment-induced complications; and design exercise experiments in preclinical models addressing simultaneously other comorbidities which may present [162].

## 4. Effects of Different Exercise Modalities (Resistance, Endurance and Concurrent Training) in Cancer Cachexia (CC)

Even though exercise training is considered a therapeutic and preventive approach for many diseases, special attention should be paid to exercise modality based on the clinical goal of the exercise intervention. Exercise modality plays a major role on specifying exercise-induced physiological adaptations and stimulating particular biological systems (e.g., cardiorespiratory, neuromuscular). Specifically, in healthy populations resistance training promotes muscle hypertrophy and muscle strength/power increase [20]. Endurance training mainly induces cardiovascular and respiratory adaptations improving respiratory and heart function [19], but also increases skeletal muscle capillary density [163,164], oxidative capacity [165,166] and mitochondria content and function [164,166]. Paradoxically, endurance training can also produce skeletal muscle hypertrophy but mainly in sedentary individuals or populations with muscle wasting, such as the elderly [167]. Concurrent training can stimulate both aerobic and anaerobic metabolic pathways, combining the beneficial adaptations induced by both resistance and endurance training [168]. Interestingly, concurrent training can be also more hypertrophic in untrained/sedentary populations compared to athletes or well-trained populations [169].

In preclinical cancer studies, as elaborated in aforementioned sections, exercise training protocols consisting of resistance training *per se* [30,57,71], endurance training *per se* [31,32,33,71,76,77,92,96,109,126] or concurrent resistance and aerobic training [58] induce beneficial adaptations in tumour-bearing mice (C26, LLC, *Apc^Min/+^,* AOM, breast-tumour bearing mice). In addition, regardless of modality, exercise training positively affects the quality of life of cancer patients [170,171]. Resistance training improves muscle strength of the upper and lower body, increases lean body mass, decreases percentage of body fat and counteracts cancer-induced increases in inflammatory factors without eliciting adverse effects in cancer patients [99,172,173]. Endurance training is also a safe and efficient method to improve aerobic capacity in cancer patients through preserving skeletal muscle mitochondrial function [56], promoting antioxidant defence system [174] and reducing fatigue symptomatology [175]. Although the majority of clinical trials emphasise the beneficial role of exercise training in cancer patients, given that specific precautions are taken (e.g., addressing comorbidities [161], individualising exercise programs [159], covering additional energetic demands [159]); a recent meta-analysis highlighted the inconsistency of data in clinical trials about the effectiveness of exercise in patients with cancer [50]. These inconsistencies can be explained by limitations (i.e., heterogeneity of participants in cancer diagnosis, cancer stages and symptoms) of the clinical trials [50].

A great number of clinical trials, as well as the guidelines by American College of Sports Medicine [59], propose a combination of both resistance and endurance training in cancer patients, since concurrent training can induce wide-ranging physical adaptations promoting both aerobic (VO_2max_ and resistance to fatigue) and anaerobic (muscle strength and function) components simultaneously without any adverse effect in cancer patients [60,176,177,178]. The phenotypic improvements of concurrent training in patients with cancer have been confirmed at the molecular level as well. Based on previous human studies, concurrent resistance and endurance training prevents cancer- and chemotherapy-induced disruptions in protein degradation and protein synthesis [78], decreases circulating inflammatory markers [98,100] and increases mitochondrial and SC content [56]. From a molecular perspective, the advantage of concurrent training in cancer patients may be related to the determinant role of mitochondria in CC. Specifically, given that cancer-induced mitochondrial impairments may lead to decrements in protein synthesis and elevated protein degradation [179], then endurance training may maximise the effects of resistance training by normalising the cancer-induced unbalanced protein turnover through improving mitochondrial health and function.

Considering the findings of exercise studies in both animals and humans with cancer, concurrent resistance and endurance training could be considered an optimal exercise modality to successfully elicit improvements in total muscle health/quality. The characteristics of concurrent training may be also crucial to produce the intended results in improving or preserving skeletal muscle mass in patients with CC. Based on literature in healthy populations, to maximise the hypertrophic effects of concurrent training, endurance and resistance elements should be separated 6–24 h [169] and if this is not feasible, the resistance exercise bout should precede the endurance exercise bout within the same session [180,181]. The endurance element should have an interval form including high-intensity bouts, ideally cycling [168,169,180,181]. More preclinical studies in tumour-bearing mice are needed to clarify the molecular and physiological mechanisms contributing to the anaerobic and aerobic improvements induced by concurrent resistance and endurance training in patients and animals with cancer. A novel, very promising and well representative model of human concurrent training, called “PoWeR”, has been recently introduced to mouse studies [182,183,184]. Briefly, “PoWeR” training consists of voluntary wheel running with progressively increased loading (from 2 g to 6 g of weight) and induces robust cardiac adaptations and skeletal muscle hypertrophy [182,183,184]. Generally, voluntary wheel running is considered a viable preclinical exercise framework that can account for the fatiguing aspects of cancer and/or chemotherapy, closing the gap between preclinical and clinical oncology studies [162]. Therefore, future preclinical studies could use the PoWeR training as a translatable murine model of concurrent training in tumour-bearing mice. An alternative to concurrent training for preclinical studies with vulnerable tumour-bearing mice could be the combination of voluntary wheel running with a novel voluntary weightlifting model for mice, which elicits squat-like activities against adjustable load during feeding [185].

To conclude, concurrent resistance and endurance training is suggested as the most beneficial modality of exercise for patients with cancer. Nevertheless, further investigation using a translatable concurrent exercise training in cancer mouse models is needed to better understand the physiological and molecular responses in cancer cachectic skeletal muscle after performing concurrent training.

## 5. Biological Sex Differences in Response to Exercise in Cancer Cachexia (CC)

Both clinical and preclinical studies in the CC field suggest biological sex differences affect muscle wasting development and progression in cancer patients and cancer mouse models [5,61,186]. Specifically, it is well established that male cancer patients present a higher prevalence of cachexia, more severe muscle mass and weight loss and greater reduction of muscle strength compared to female cancer patients [61]. In addition, animal studies have suggested the cellular and molecular mechanisms contributing to CC, as well as the pace of CC development and progression, are different between male and female cachectic mice [15,61,187]. Specifically, male *Apc^Min/+^* mice are sensitive to inflammation (IL-6)-mediated cachexia [6], while female *Apc^Min/+^* mice undergo cachexia progression IL-6-independently [187]. Similarly, female LLC tumour-bearing mice present multiple protections in both metabolic and contractile skeletal muscle function in the early stages of tumour development [15], while male LLC tumour-bearing mice present many metabolic perturbations before the onset of cachexia [7,14]. Therefore, female tumour-bearing mice may have a stronger defence system protecting them from tumour-induced muscle degeneration [15].

Generally speaking, female muscles are more fatigue-resistant and appear to exhibit enhanced mitochondrial quality compared to male muscles [61,62]. However, males exhibit greater SC content [188] and increased mRNA expression of MyoD and myogenin [189], two factors associated with SC proliferation and differentiation and potentially skeletal muscle hypertrophic and regenerative potential. Considering the biological sex dissimilarities observed in the skeletal muscle, different skeletal muscle adaptations in response to a metabolic stimulus, such as exercise training, may be induced in males and females. Indeed, a previous human study based on young and healthy adults concluded that males present greater muscle protein synthesis and mitochondrial biogenesis than females in response to 3-weeks of sprint interval training [190]. However, when low-to-moderate continuous endurance training (cycling at 60% of VO_2max_, 5 days/week, 7 weeks) is performed, no difference is observed in skeletal muscle enzyme oxidative potential between males and females [191]. Furthermore, acute high-intensity resistance training combined with post-exercise whey protein ingestion increases myofibrillar protein synthesis and p70S6K1 phosphorylation similarly in males and females, although the postexercise testosterone responses in females are 45-fold lower than males [192]. Conversely, compared to male, female skeletal muscle is more resistant to fatigue for a task of the same relative intensity [62,193,194]. Females present a lower degree of contractile impairment and faster rate of recovery following exercise with intermittent isometric knee extensions [194].

To our knowledge, no study has directly investigated biological sex differences in response to exercise in cancer patients or tumour-bearing mice. However, an indirect and observational comparison between preclinical studies with different biological sex suggests exercise-induced adaptations in skeletal muscle are similar in female and male tumour-bearing mice [57,64]. Specifically, a representative resistance training for animals involving lower limb contractions through high-frequency electric stimulation for 2-weeks, increases weight and size of Type IIa and IIb fibres in the TA muscle, improves myofibrillar protein synthesis, as well as attenuates cachexia-induced AMPK activity and reduction of skeletal muscle oxidative capacity similarly in *Apc^Min/+^* female [64] and male *Apc^Min/+^* mice [57]. In addition, 2-week resistance training in the form of high-frequency electric stimulation increases EDL weight and protein content in female C26 mice [195]. Although no study has investigated the effects of resistance training *per se* in male C26 mice, previous studies have shown that even endurance training *per se* or concurrent endurance and resistance training can induce beneficial skeletal muscle adaptations in male C26 mice preventing cancer-induced muscle wasting and loss of muscle strength [58,76]. Also, in both male [76] and female [32] C26 mice physical activity via wheel running suppresses the protein expression of the autophagic markers LC3B and p62.

The beneficial effects of exercise training in cancer patients appear similar between males and females [196]. Specifically, clinical studies that recruited either female [197,198] or male [199,200] cancer patients suggest exercise is an efficient therapeutic intervention for preserving muscle wasting and muscle strength loss. However, similarly with preclinical cancer studies, no clinical study has directly investigated biological sex differences in exercise-induced adaptations in cancer patients. Future preclinical exercise studies in CC mouse models should be focused on assessing potential biological sex differences in skeletal muscle adaptations in response to exercise training. Discovering potential differences in exercise-induced skeletal muscle adaptations between male and female tumour-bearing mice could guide personalised exercise training protocols based on biological sex, thereby optimising the effects of exercise training in both male and female cancer patients.

## 6. Conclusions

There is now a large body of evidence suggesting exercise training as a beneficial therapeutic approach for decelerating the development and progression of cancer-induced atrophy. Preclinical and clinical studies have shown that exercise training, regardless of modality, beneficially regulates the molecular and cellular mechanisms that contribute to CC pathology. Protein turnover, inflammatory signalling, mitochondrial function and satellite cell number can be advantageously controlled by exercise training in animals and patients with cancer. Given the majority of cancer patients present at least one comorbidity, exercise training could elicit adverse events in these vulnerable individuals. For this reason, meticulous precautions (e.g., low-intensity exercise training protocols under constant supervision by specialists) should be taken while exercise is performed. In many instances, a complete address of comorbidities (e.g., haemoglobin back or close to the normal levels) may be essential before cancer patients begin an exercise training program. Fatigue of cancer patients related to either comorbidities (e.g., anaemia) or radiation therapies/chemotherapies should be also taken into account for the exercise training program prescription to avoid exercise adverse effects in these vulnerable patients.

Based on findings from preclinical and clinicals studies, as well as the guidelines of American College of Sports Medicine for cancer patients, concurrent resistance and endurance training is considered the most advantageous exercise modality for cancer patients. Concurrent training simultaneously improves elements of both aerobic and anaerobic metabolism. Nevertheless, further investigation is needed to understand the molecular mechanisms activated through concurrent training and contributing to improving skeletal muscle health of patients with CC. A novel voluntary wheel running protocol for mice, called PoWeR training [182,183,184], could be a representative exercise mode of concurrent resistance and endurance training. Generally, voluntary wheel running in cancer mice is considered a translatable exercise mode since it accounts for the fatiguing aspects induced by comorbidities or side effects of cancer treatments. Consequently, future preclinical studies using the PoWeR exercise training protocol in cancer mice could explain many aspects of mechanisms underlying the deceleration of cancer-induced atrophy after performing a concurrent training protocol, which is appropriate even if comorbidities are presented along with cancer.

Finally, when comparing observationally the beneficial exercise-induced adaptations between male and female patients or mice with cancer, both biological sexes seem to respond similarly to exercise stimuli. However, neither preclinical nor clinical studies in the cancer field have investigated directly the potential biological sex differences on the skeletal muscle in response to exercise training in the tumour-bearing state. Given that dissimilarities are observed in CC development and progression between females and males, future investigation is needed to understand if the biological sex in cancer patients plays a crucial role on the exercise-induced adaptations. The Figure 2 summarises all of the aforementioned conclusions of this review.

## Figures and Tables

**Figure 2 cancers-14-02512-f002:**
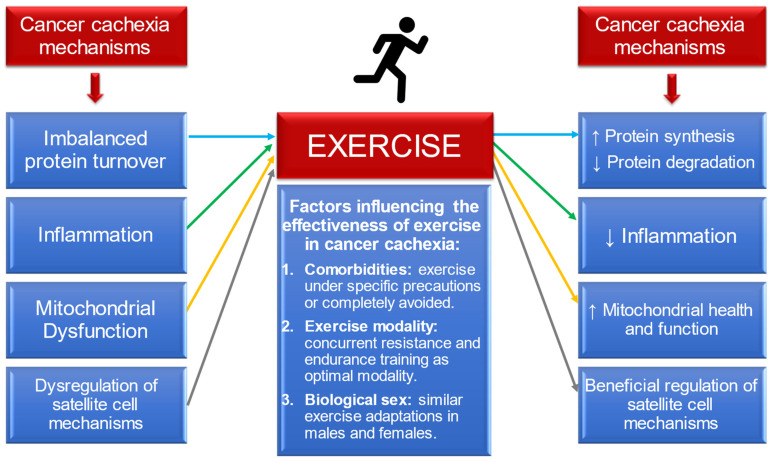
Schematic summary of the review: effects of exercise training in the major mechanisms of cancer cachexia (protein turnover, inflammation, mitochondrial function and satellite cell regulation); and the influence of cancer comorbidities, exercise modality and biological sex in the effectiveness of exercise training in cancer cachexia.

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
