# Peer review of "Exercise Counteracts the Deleterious Effects of Cancer Cachexia"

_cancers, 2022, doi:10.3390/cancers14102512_

Round 1
Reviewer 1 Report
The authors have put together a very comprehensive review on how various phenotypes and mechanisms associated with cachexia can get modulated by exercise. Also, they nicely put together the various considerations that are important to know when exercise-related treatment options are provided to people dealing with the debilitating condition of cachexia. The table and the figure is a good summary of the textual content of the review.
Some suggestions:
- Since there is a lot of focus on mitochondrial health it would be good to include information from https://www.nature.com/articles/s41598-019-49010-6 too.
- For the section on exercise modalities will be good to include information form this new article https://link.springer.com/article/10.1007/s00432-022-03927-0.
- The authors should include a table of any ongoing clinical trials for exercise based interventions.
- A figure to represent the different mechanisms of cachexia (not just a text based figure like the one that authors have) to show the signaling and how exercise affects those would be a nice visual representation for the readers.
- Please cite some more recent articles- e.g- https://pubmed.ncbi.nlm.nih.gov/32954861/
- Spelling error of satellite -2.4. Sattelite cells and myogenic regulatory factors in response to exercise.
Author Response
Reviewer 1: The authors have put together a very comprehensive review on how various phenotypes and mechanisms associated with cachexia can get modulated by exercise. Also, they nicely put together the various considerations that are important to know when exercise-related treatment options are provided to people dealing with the debilitating condition of cachexia. The table and the figure is a good summary of the textual content of the review.
Response: Thank you for the kind comments regarding our work.
Some suggestions:
1. Since there is a lot of focus on mitochondrial health it would be good to include information from https://www.nature.com/articles/s41598-019-49010-6 too.
Response: Thank you for your suggestion. The article is now included in the section “Mitochondrial function and health in response to exercise”, line 308-310, ref. 111 (“Mitochondrial function and health are also improved with exercise training in patients and rodents with cancer”). Specifically, we believe these data help to corroborate findings currently referenced in this location while broadening the impacts of these studies across rodent models (mouse and rat).
2. For the section on exercise modalities will be good to include information form this new article https://link.springer.com/article/10.1007/s00432-022-03927-0.
Response: Thank you for your suggestion. The article is now included in the respect section, line 540-543, ref. 174 (“Resistance training improves muscle strength of the upper and lower body, increases lean body mass, decreases percentage of body fat and counteracts cancer-induced increases in inflammatory factors without eliciting adverse effects in cancer patients”).
3. The authors should include a table of any ongoing clinical trials for exercise based interventions.
Response: This is a great suggestion. Because of high number of the ongoing clinical trials, the Table has been included as a Supplementary material (see Supplementary Table 1).
4. A figure to represent the different mechanisms of cachexia (not just a text based figure like the one that authors have) to show the signaling and how exercise affects those would be a nice visual representation for the readers.
Response: Thank you for this comment, we have added the new Figure 1 to the revised manuscript as recommended (line 435).
5. Please cite some more recent articles- e.g- https://pubmed.ncbi.nlm.nih.gov/32954861/
Response: More recent articles (ref. 36, 42, 50, 51, 160, 174, 175), including your suggested paper (ref.42, line 91-93: “For this reason, the majority of pre-clinical studies start the exercise intervention before or just after the inoculation of cancer cells to succeed exercise-induced adaptations42”), have been added to the revised text.
6. Spelling error of satellite -2.4. Sattelite cells and myogenic regulatory factors in response to exercise.
Response: Thank you for observing this typo. It is now corrected.
Reviewer 2 Report
Overview of exercise and specific mechanisms related to cachexia in oncology. Introduction: has a great indepth description of CC at the physical/chemical levels.
Overall very detailed in depth article which provides a lot of review and data which can be useful for upcoming design of future studies.
Suggestions:
Can remove Lines 42/43/44 as readers of this journal understand this already. Can start with line ‘Cancer cachexia’ as this is what really introduces this manuscript.
All sections very lengthy and detailed. Would consider simplifying to allow readers to read thru and have a higher yield from this paper.
Author Response
Overview of exercise and specific mechanisms related to cachexia in oncology. Introduction: has a great indepth description of CC at the physical/chemical levels.
Overall very detailed in depth article which provides a lot of review and data which can be useful for upcoming design of future studies.
Response: Thank you for the kind comments regarding our work.
Suggestions:
Can remove Lines 42/43/44 as readers of this journal understand this already. Can start with line ‘Cancer cachexia’ as this is what really introduces this manuscript.
Response: Thank you for your suggestion. The respective lines have been deleted.
All sections very lengthy and detailed. Would consider simplifying to allow readers to read thru and have a higher yield from this paper.
Response: Thank you for this comment. In our revision to accommodate the goal of this request we have added a new figure (Figure 1). The Figure 1 is now included in the manuscript to simplify the text. However, to concomitantly maintain the depth of the review and provide the reader with critical information we have opted to maintain the detailed information. We feel this provides the reader with the best possible information about the discussed subject.
Reviewer 3 Report
The review entitled “Exercise counteracts the deleterious effects of cancer cachexia” examines an important topic. The authors want to focus on the skeletal muscle alterations in cachectic cancer patients in a context of exercise and to identify the main mechanisms. The thematic of this review is original and of great interest, in order to improve the management care of cachectic patients. However, two major points remain to be clarified and the paper should be better organized, to gain efficiency and clarity.
1 / The authors have to read/cite the different review below in order to implement the section about the impact of exercise training in cancer cachexia management. It is important to discuss in the review that this topic is poorly documented in patients. In its current form, the authors suggest that physical activity can slow the progression of cancer cachexia. However, the data in patients are very inconsistent.
Allan, Jessica, Linda A. Buss, Nick Draper, et Margaret J. Currie. « Exercise in People With Cancer: A Spotlight on Energy Regulation and Cachexia ». Frontiers in Physiology 13 (2022): 836804. https://doi.org/10.3389/fphys.2022.836804.
Aquila, Giorgio, Andrea David Re Cecconi, Jeffrey J. Brault, Oscar Corli, et Rosanna Piccirillo. « Nutraceuticals and Exercise against Muscle Wasting during Cancer Cachexia ». Cells 9, no 12 (24 novembre 2020): E2536. https://doi.org/10.3390/cells9122536.
« Cancer cachexia: rationale for the MENAC (Multimodal-Exercise, Nutrition and Anti-inflammatory medication for Cachexia) trial - PubMed ». Consulté le 27 avril 2022. https://pubmed.ncbi.nlm.nih.gov/29440149/.
Delphan, Mahmoud, Neda Delfan, Daniel West, et Maryam Delfan. « Exercise Protocols: The Gap between Preclinical and Clinical Exercise Oncology Studies ». Metabolism Open 13 (mars 2022): 100165. https://doi.org/10.1016/j.metop.2022.100165.
Grande, Antonio Jose, Valter Silva, Larissa Sawaris Neto, João Pedro Teixeira Basmage, Maria S. Peccin, et Matthew Maddocks. « Exercise for Cancer Cachexia in Adults ». The Cochrane Database of Systematic Reviews 3 (18 mars 2021): CD010804. https://doi.org/10.1002/14651858.CD010804.pub3.
Leal, Luana G., Magno A. Lopes, Sidney B. Peres, et Miguel L. Batista. « Exercise Training as Therapeutic Approach in Cancer Cachexia: A Review of Potential Anti-Inflammatory Effect on Muscle Wasting ». Frontiers in Physiology 11 (2020): 570170. https://doi.org/10.3389/fphys.2020.570170.
Liguori, Sara. « Is Exercise Effective and Safe for Cancer Cachexia in Adults? - A Cochrane Review Summary with Commentary ». American Journal of Physical Medicine & Rehabilitation, 19 janvier 2022. https://doi.org/10.1097/PHM.0000000000001973.
Mader, Theresa, Thomas Chaillou, Estela Santos Alves, Baptiste Jude, Arthur J. Cheng, Ellinor Kenne, Sara Mijwel, et al. « Exercise Reduces Intramuscular Stress and Counteracts Muscle Weakness in Mice with Breast Cancer ». Journal of Cachexia, Sarcopenia and Muscle 13, no 2 (avril 2022): 1151‑63. https://doi.org/10.1002/jcsm.12944.
Mavropalias, Georgios, Marc Sim, Dennis R. Taaffe, Daniel A. Galvão, Nigel Spry, William J. Kraemer, Keijo Häkkinen, et Robert U. Newton. « Exercise Medicine for Cancer Cachexia: Targeted Exercise to Counteract Mechanisms and Treatment Side Effects ». Journal of Cancer Research and Clinical Oncology, 27 janvier 2022. https://doi.org/10.1007/s00432-022-03927-0.
Niels, Timo, Annika Tomanek, Nils Freitag, et Moritz Schumann. « Can Exercise Counteract Cancer Cachexia? A Systematic Literature Review and Meta-Analysis ». Integrative Cancer Therapies 19 (décembre 2020): 1534735420940414. https://doi.org/10.1177/1534735420940414.
2 / When dealing with data obtained in preclinical models, it is important to distinguish between animal models where exercise is performed after cachexia has occurred and those where exercise is started before. Indeed, in the title, the authors mention that exercise can counteract the deleterious effects of cancer cachexia. However, the vast majority of the articles cited show that in an animal model of cancer that leads to cachexia, exercise started before or just after the injection of cancer cells delays the development of cachexia. It is therefore necessary to distinguish between development and progression. In cancer patients, physical activity can delay the development of cachexia, but in the patient with established cachexia, what is the result? The authors' message is not clear. It is important to clarify this point and to rework the article in the light of this argument.
Author Response
The review entitled “Exercise counteracts the deleterious effects of cancer cachexia” examines an important topic. The authors want to focus on the skeletal muscle alterations in cachectic cancer patients in a context of exercise and to identify the main mechanisms. The thematic of this review is original and of great interest, in order to improve the management care of cachectic patients. However, two major points remain to be clarified and the paper should be better organized, to gain efficiency and clarity.
1 / The authors have to read/cite the different review below in order to implement the section about the impact of exercise training in cancer cachexia management. It is important to discuss in the review that this topic is poorly documented in patients. In its current form, the authors suggest that physical activity can slow the progression of cancer cachexia. However, the data in patients are very inconsistent.
Allan, Jessica, Linda A. Buss, Nick Draper, et Margaret J. Currie. « Exercise in People With Cancer: A Spotlight on Energy Regulation and Cachexia ». Frontiers in Physiology 13 (2022): 836804. https://doi.org/10.3389/fphys.2022.836804.
Aquila, Giorgio, Andrea David Re Cecconi, Jeffrey J. Brault, Oscar Corli, et Rosanna Piccirillo. « Nutraceuticals and Exercise against Muscle Wasting during Cancer Cachexia ». Cells 9, no 12 (24 novembre 2020): E2536. https://doi.org/10.3390/cells9122536.
« Cancer cachexia: rationale for the MENAC (Multimodal-Exercise, Nutrition and Anti-inflammatory medication for Cachexia) trial - PubMed ». Consulté le 27 avril 2022. https://pubmed.ncbi.nlm.nih.gov/29440149/.
Delphan, Mahmoud, Neda Delfan, Daniel West, et Maryam Delfan. « Exercise Protocols: The Gap between Preclinical and Clinical Exercise Oncology Studies ». Metabolism Open 13 (mars 2022): 100165. https://doi.org/10.1016/j.metop.2022.100165.
Grande, Antonio Jose, Valter Silva, Larissa Sawaris Neto, João Pedro Teixeira Basmage, Maria S. Peccin, et Matthew Maddocks. « Exercise for Cancer Cachexia in Adults ». The Cochrane Database of Systematic Reviews 3 (18 mars 2021): CD010804. https://doi.org/10.1002/14651858.CD010804.pub3.
Leal, Luana G., Magno A. Lopes, Sidney B. Peres, et Miguel L. Batista. « Exercise Training as Therapeutic Approach in Cancer Cachexia: A Review of Potential Anti-Inflammatory Effect on Muscle Wasting ». Frontiers in Physiology 11 (2020): 570170. https://doi.org/10.3389/fphys.2020.570170.
Liguori, Sara. « Is Exercise Effective and Safe for Cancer Cachexia in Adults? - A Cochrane Review Summary with Commentary ». American Journal of Physical Medicine & Rehabilitation, 19 janvier 2022. https://doi.org/10.1097/PHM.0000000000001973.
Mader, Theresa, Thomas Chaillou, Estela Santos Alves, Baptiste Jude, Arthur J. Cheng, Ellinor Kenne, Sara Mijwel, et al. « Exercise Reduces Intramuscular Stress and Counteracts Muscle Weakness in Mice with Breast Cancer ». Journal of Cachexia, Sarcopenia and Muscle 13, no 2 (avril 2022): 1151‑63. https://doi.org/10.1002/jcsm.12944.
Mavropalias, Georgios, Marc Sim, Dennis R. Taaffe, Daniel A. Galvão, Nigel Spry, William J. Kraemer, Keijo Häkkinen, et Robert U. Newton. « Exercise Medicine for Cancer Cachexia: Targeted Exercise to Counteract Mechanisms and Treatment Side Effects ». Journal of Cancer Research and Clinical Oncology, 27 janvier 2022. https://doi.org/10.1007/s00432-022-03927-0.
Niels, Timo, Annika Tomanek, Nils Freitag, et Moritz Schumann. « Can Exercise Counteract Cancer Cachexia? A Systematic Literature Review and Meta-Analysis ». Integrative Cancer Therapies 19 (décembre 2020): 1534735420940414. https://doi.org/10.1177/1534735420940414.
Response: Thank you for your recommendations and providing the list of recent publications in the field. These articles are now included in our manuscript (ref. 36, 42, 50, 51, 160, 163, 174, 175) and the inconsistency of data in clinical trials with cancer patient has been also commended in our text (line 114-119: “Given the heterogeneity observed in cancer patients (i.e. different cancer diagnoses, cancer stages and symptoms) in exercise clinical trials, further works are necessary to evaluate exercise training as a monotherapy or combinatorial therapy throughout all phases of cancer, including the phases of CC development and progression 42, 50, 51. To this point, ongoing clinical trials with exercise-based interventions in cancer patients are presented in the new Supplementary Table 1”; line 545-552: “Although the majority of clinical trials emphasise the beneficial role of exercise training in cancer patients, given that specific precautions are taken (e.g., addressing comorbidities162, individualising exercise programs160, covering additional energetic demands160); a recent meta-analysis highlighted the inconsistency of data in clinical trials about the effectiveness of exercise in patients with cancer50. These inconsistencies can be explained by limitations (i.e., heterogeneity of participants in cancer diagnosis, cancer stages and symptoms) of the clinical trials 50).
2 / When dealing with data obtained in preclinical models, it is important to distinguish between animal models where exercise is performed after cachexia has occurred and those where exercise is started before. Indeed, in the title, the authors mention that exercise can counteract the deleterious effects of cancer cachexia. However, the vast majority of the articles cited show that in an animal model of cancer that leads to cachexia, exercise started before or just after the injection of cancer cells delays the development of cachexia. It is therefore necessary to distinguish between development and progression. In cancer patients, physical activity can delay the development of cachexia, but in the patient with established cachexia, what is the result? The authors' message is not clear. It is important to clarify this point and to rework the article in the light of this argument.
Response:
Thank you for highlighting this. The designation as to whether this research, or clinical work, is preventive (prior to onset of cachexia) or based upon reversal (correctly existent cachexia) is of course critical. Based on this we have commented the limitations of exercise pre-clinical and clinical studies in cancer patients and/or mice: i) line 89-105: “A major unavoidable limitation of pre-clinical studies in CC mouse models is the short timeline (3-4 weeks) the mice have after the inoculation of cancer cells. For this reason, the majority of pre-clinical studies begin the exercise intervention before or just after the inoculation of cancer cells to succeed exercise-induced adaptations. This timepoint is undoubtedly earlier than the development and progression of CC (around 3-4 weeks after cancer cells inoculation). For this reason, most pre-clinical studies are designed with prevention of cachexia as the primary goal leaving a relative dearth in data regarding reversal of existent cachexia”; ii) line 109-119: “Exercise training can be also advantageous as a pre-operative intervention in cancer patients 47-49. Specifically, exercise not only improves the physical capacity and function of cancer patients 47, 48, but also ameliorates postoperative recovery 47 and may have a protective direct effect on tumour growth through altering the gene expression of immunity and inflammation “key” factors 49. Given the heterogeneity observed in cancer patients (i.e. different cancer diagnoses, cancer stages and symptoms) in exercise clinical trials, further works are necessary to evaluate exercise training as a monotherapy or combinatorial therapy throughout all phases of cancer, including the phases of CC development and progression 42, 50, 51. To this point, ongoing clinical trials with exercise-based interventions in cancer patients are presented in the new Supplementary Table 1”.
Reviewer 4 Report
This review focuses on the effect of exercise on the major mechanisms underlying CC, giving an accurate overview of published studies either preclinical in animals and clinical in cancer patients, also evaluating possible gender-related differences, that have never been analysed so far. The review is very well addressed and complete, it gives important warnings on the risks for exercise in cancer patients, also providing potential protocols of combined strategies of training. No major criticisms.
Minor points:
- there is a growing body of knowledge on the positive effects of exercise as a pre-operative intervention in cancer patients, not only in terms of postoperative outcomes, but also of disease progression. Even if this topic is probably far from the main focus of the review, it would be important to include a comment on this aspect in the Introduction, to strengthen the crucial role of exercise in cancer-related conditions.
- Table 1 formatting is not adequate for an easy reading and comprehension. It would be better to justify the text or put it into columns.
- On the title of paragraph 2.4 the word “satellite” is misspelled.
Author Response
This review focuses on the effect of exercise on the major mechanisms underlying CC, giving an accurate overview of published studies either preclinical in animals and clinical in cancer patients, also evaluating possible gender-related differences, that have never been analysed so far. The review is very well addressed and complete, it gives important warnings on the risks for exercise in cancer patients, also providing potential protocols of combined strategies of training. No major criticisms.
Response: Thank you for the kind comments regarding our work.
Minor points:
There is a growing body of knowledge on the positive effects of exercise as a pre-operative intervention in cancer patients, not only in terms of postoperative outcomes, but also of disease progression. Even if this topic is probably far from the main focus of the review, it would be important to include a comment on this aspect in the Introduction, to strengthen the crucial role of exercise in cancer-related conditions.
Response: Thank you for your recommendation. Exercise training as a pre-operative intervention has been commented in the revised Introduction (lines 109-114: “Exercise training can be also advantageous as a pre-operative intervention in cancer patients 47-49. Specifically, exercise not only improves the physical capacity and function of cancer patients 47, 48, but also ameliorates postoperative recovery 47 and may have a protective direct effect on tumour growth through altering the gene expression of immunity and inflammation “key” factors 49”).
Table 1 formatting is not adequate for an easy reading and comprehension. It would be better to justify the text or put it into columns.
Response: The Table 1 has been reformatted for easier reading and comprehension.
On the title of paragraph 2.4 the word “satellite” is misspelled.
Response: Thank you for observing this typo. It is now corrected.
Round 2
Reviewer 3 Report
The reviewer thanks the authors for making the requested changes.